# Topological magneto-optical effect from skyrmion lattice

Yoshihiro D. Kato [1], Yoshihiro Okamura [1] ✉, Max Hirschberger [1,2], Yoshinori Tokura [1,2,3] & Youtarou Takahashi [1,2] ✉

The magnetic skyrmion is a spin-swirling topological object characterized by its nontrivial winding number, holding potential for next-generation spin-tronic devices. While optical readout has become increasingly important towards the high integration and ultrafast operation of those devices, the optical response of skyrmions has remained elusive. Here, we show the magneto-optical Kerr effect (MOKE) induced by the skyrmion formation, i.e., topological MOKE, in $Gd_2PdSi_3$. The significantly enhanced optical rotation found in the skyrmion phase demonstrates the emergence of topological MOKE, exemplifying the light-skyrmion interaction arising from the emergent gauge field. This gauge field in momentum space causes a dramatic reconstruction of the electronic band structure, giving rise to magneto-optical activity ranging up to the sub-eV region. The present findings pave a way for photonic technology based on skyrmionics.

Since its first discovery in a chiral magnet, the magnetic skyrmion, a nanometric quasiparticle composed of full-solid-angle oriented spin moments, has attracted much attention because of its potential for high density and low power consumption memory/logic devices[1-8]. The topological nature of each skyrmion, which is derived from the directions of constituent spins wrapping the unit sphere, gives rise to the robust stability and high electric-current controllability of the skyrmion particle, being useful as the information carrier[3,9,10]. This particular spin arrangement also hosts the quantized scalar spin chirality, generating a gauge field originating from the Berry phase. As a result, a fictitious magnetic field acts on the electronic system, often referred to as an emergent magnetic field, due to the skyrmion formation. For conduction electrons, this emergent magnetic field induces topological transport phenomena as exemplified by the topological Hall effect (THE) and topological Nernst effect[9,11-14], which can be exploited for electrical readout of skyrmions. In recent years, much effort has been devoted to the exploration of new materials hosting small skyrmions[15-18] and to the detailed elucidation of skyrmion dynamics such as the current-induced motion and creation/annihilation processes[19-21]. These fundamental studies enable further advances towards the realization of higher density devices and their operation. On the other hand, these findings warrant the development of more sophisticated readout schemes, beyond simple transport experiments (THE), which are capable of sensitive, functional, and high-speed response.

In this context, the magneto-optical effect, a light polarization rotation under breaking of time-reversal symmetry, is a promising candidate. It has been used as a local, fast, and contactless probe of magnetic domains[22]. The magnitude of the magneto-optical effect is usually proportional to the magnetization ($M$), and thus it is believed to be less sensitive to the emergence of skyrmions accompanying weak change in the net $M$[23]. Meanwhile, some recent theories predict the so-called topological magneto-optical effect induced by the formation of noncoplanar spin structures with finite scalar spin chirality[24]. This mechanism is essentially distinct from the conventional $M$-induced magneto-optical effect governed by the interplay between band exchange splitting and relativistic spin-orbit coupling[25,26]; the spin-orbit coupling is not a prerequisite for the topological magneto-optical effect[24,27]. Thus, the emergent magnetic field arising from skyrmion formation potentially gives rise to the topological magneto-optical effect sensitive to the existence of skyrmion, analogous to the THE[24,27], which can be exploited for the optical detection of skyrmions.

[1]Department of Applied Physics and Quantum Phase Electronics Center, University of Tokyo, Tokyo 113-8656, Japan. [2]RIKEN Center for Emergent Matter Science (CEMS), Wako 351-0198, Japan. [3]Tokyo College, University of Tokyo, Tokyo 113-8656, Japan. ✉e-mail: okamura@ap.t.u-tokyo.ac.jp; youtarou-takahashi@ap.t.u-tokyo.ac.jp

However, such an optical response driven by skyrmion formation has yet to be elucidated.

Here, we report on the topological MOKE arising from the skyrmion lattice (SkL) in the centrosymmetric rare-earth intermetallic compound $Gd_2PdSi_3$. By using broadband magneto-optical spectroscopy, we observe a largely enhanced MOKE in the sub-eV region due to SkL formation, evidencing the existence of topological MOKE and a significant change in Bloch electron wavefunctions constituting band structure by the emergent magnetic field. Such a reconstruction of electronic bands is found to contribute to THE, providing a comprehensive understanding of emergent electrodynamics over an extended energy scale.

## Results

The rare-earth intermetallic $Gd_2PdSi_3$ crystallizes in the $AlB_2$-type structure; a triangular lattice of Gd atoms sandwiches a nonmagnetic honeycomb-lattice layer composed of Pd and Si atoms (Fig. 1a)[28]. Long-range magnetic order of the Gd 4f moments is stabilized below 20 K[14,15,29]. Because of frustrated magnetic interactions on the triangular network of Gd moments, this material shows a rich magnetic phase diagram including modulated spin structures with short magnetic periods (Fig. 1a, b). The SkL appears under moderate magnetic field parallel to the c axis, in between two incommensurate magnetic phases without net emergent magnetic field: spiral-like IC-1 and fan-like IC-2 states (Supplementary Fig. 1)[14,15,29]. Each skyrmion is as small as a few nanometers in diameter, much smaller than the size of such textures in conventional chiral magnets. Since one skyrmion provides a single flux quantum, the high-density SkL generates exceedingly strong emergent magnetic fields, leading to enhanced topological transport

phenomena; the Hall conductivity is steeply enhanced in the SkL phase due to the giant THE (Fig. 1c), while $M$ shows a monotonic increase, with a step-like anomaly at each phase boundary[14,15].

To pursue the topological magneto-optical effect, we measured broadband MOKE spectra, i.e., the polarization rotation of reflected light from the sample surface, under magnetic fields parallel to the c-axis. Figure 1d, e shows the magnetic field dependence of the Kerr rotation angle $\theta_K(\omega)$ (0.04–0.08 eV and 0.1–0.8 eV) and Kerr ellipticity $\eta_K(\omega)$ (0.1–0.8 eV) at 8.4 K (see also Methods). The overall magnitude of magneto-optical responses tends to grow with increasing magnetic field, while a dramatic change of the MOKE spectra is observed upon entering the SkL phase. The IC-1 and IC-2 phases with no net emergent magnetic field show common spectral features, i.e., a broad peak structure centered at 0.6 eV in $\theta_K(\omega)$ and a negative peak at 0.45 eV in $\eta_K(\omega)$ (green, blue, and purple curves in Fig. 1d, e). In addition to these resonance structures, the hump around 0.3 eV in $\theta_K(\omega)$ and the dip at 0.2 eV in $\eta_K(\omega)$ appear only in the SkL phase (red and orange curves), providing a clear fingerprint for the emergence of the topological MOKE. We note that these spectral changes are also observed in the temperature dependence (Supplementary Fig. 2). These energy-dependent anomalies in the MOKE spectra are further corroborated by constant energy scans along magnetic field (Fig. 2a); $\theta_K(\omega)$ shows a steep enhancement at 0.3 eV and decrease at 0.15 eV due to skyrmion formation, which is impossible to explain by the conventional MOKE proportional to $M$.

Since the observed magnetic-field dependence of $\theta_K(\omega)$ resembles the anomaly in the d.c. Hall effect shown in Fig. 1c, we aim to extract the topological MOKE by subtracting the $M$-induced MOKE $\theta_K^M(\omega)$, in analogy to the conventional analysis for the THE. Namely, we define

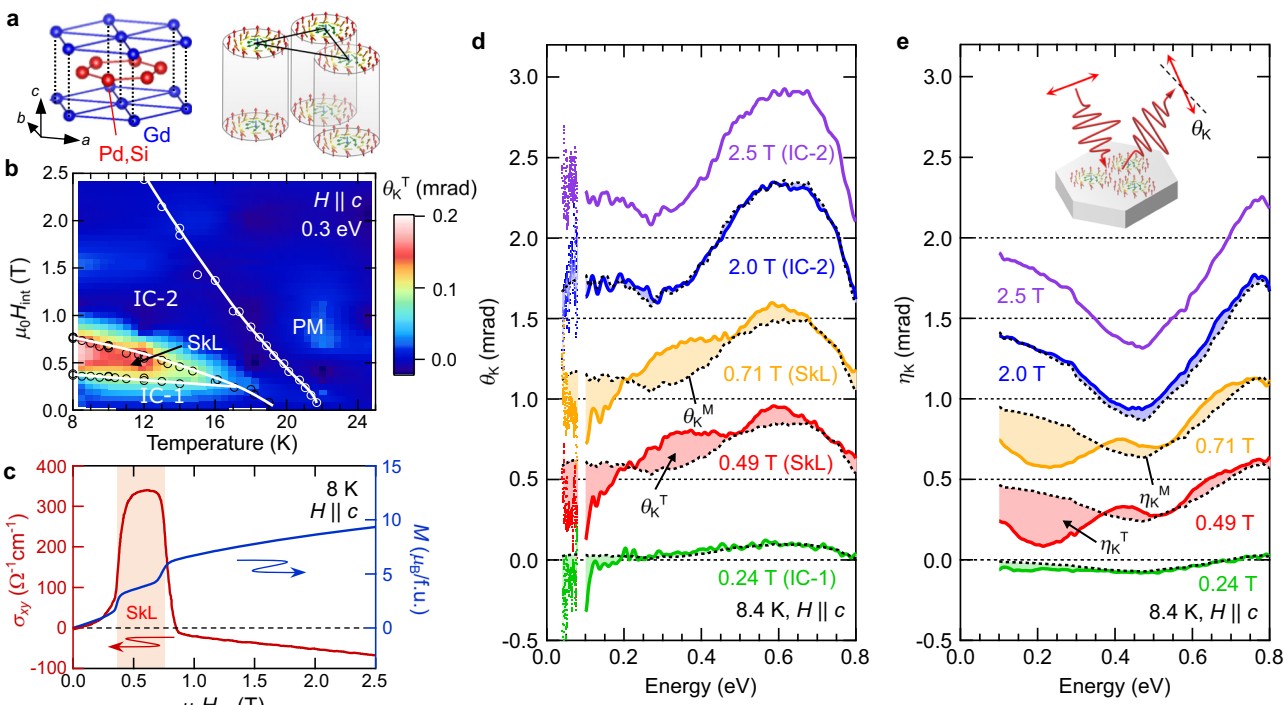

**Fig. 1 | Basic properties and magneto-optical Kerr effect of $Gd_2PdSi_3$. a** The $AlB_2$-type crystal structure of $Gd_2PdSi_3$ (left panel) and the schematic illustration of the triangular skyrmion lattice (SkL) (right panel). **b** Magnetic phase diagram with magnetic field ($H$) parallel to the c axis and a contour map of topological Kerr rotation angle $\theta_K^T$ at 0.3 eV. IC-1 and IC-2 represent incommensurate spin-state phases (see Supplementary Fig. 1), and PM represents the paramagnetic phase. The open circles represent phase boundaries determined by magnetization measurements. **c** Magnetic-field dependence of the d.c. Hall conductivity $\sigma_{xy}$ (left axis) and magnetization $M$ (right axis) for $H \parallel c$ at 8 K. $H_{int}$ represents the internal magnetic field, considering the demagnetization effect. The red shaded area denotes the SkL phase. **d, e** Magnetic-field dependence of the magneto-optical Kerr (**d**) rotation angle $\theta_K$ and (**e**) ellipticity $\eta_K$ for $H \parallel c$ at 8.4 K. The data are shifted by vertical offsets of 0.5 mrad. The black dotted curves in (**d**) and (**e**) denote the conventional $M$-linear MOKE $\theta_K^M$ and $\eta_K^M$; the shaded areas in (**d**) and (**e**) represent the topological components $\theta_K^T$ and $\eta_K^T$ (see main text and Supplementary Fig. 3 for their definition). The inset in (**e**) shows the schematic illustration of the MOKE measurement geometry.

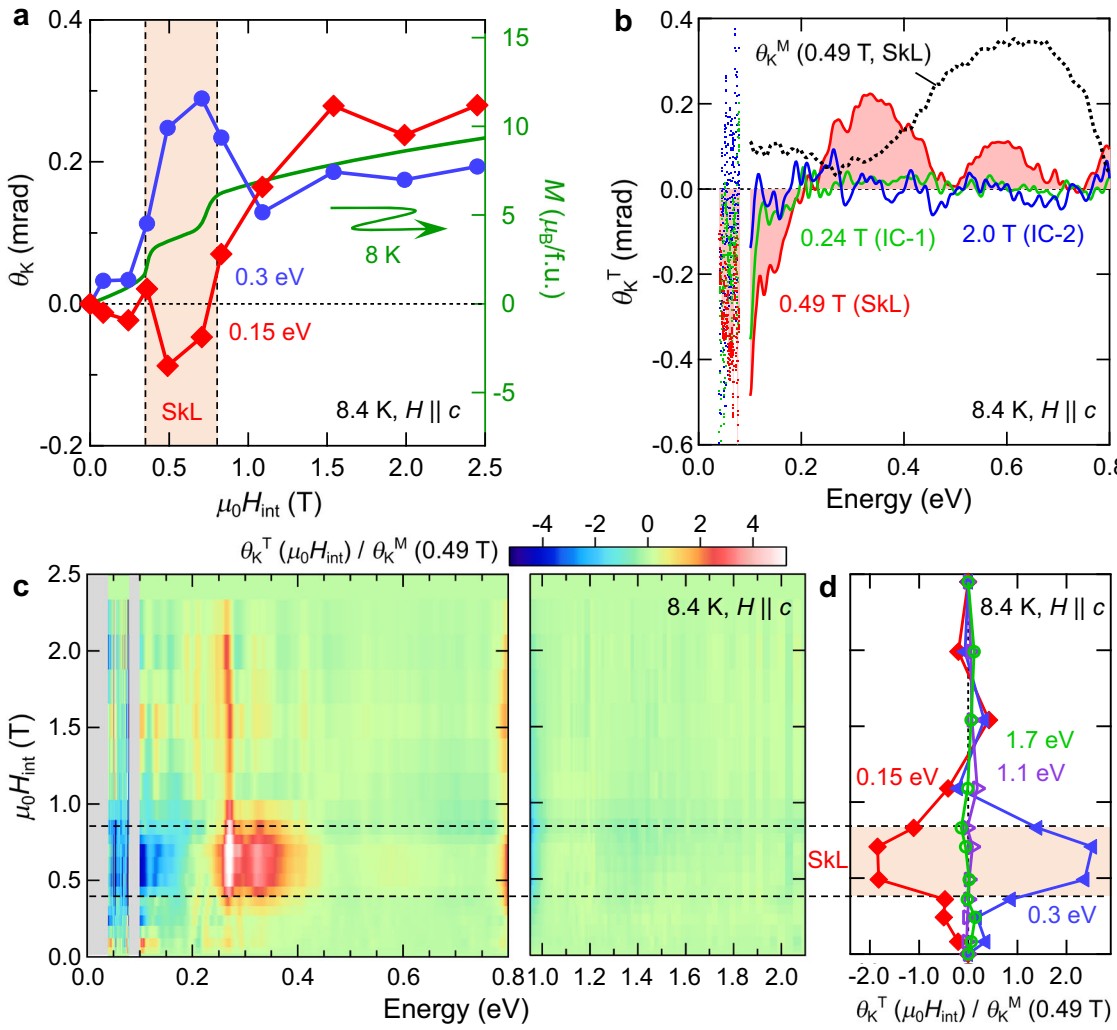

**Fig. 2 | Topological magneto-optical effect from SkL. a** Magnetic-field dependence of the Kerr rotation $\theta_K$ at 0.15 eV (red) and at 0.3 eV (blue) at 8.4 K, and of the bulk magnetization $M$ (green curve) at 8 K. The dotted vertical lines represent the phase boundaries of the SkL phase. **b** Topological Kerr rotation $\theta_K^T$ at 0.24 T (IC-1 phase, green), at 0.49 T (SkL phase, red), and at 2.0 T (IC-2 phase, blue). The black dotted curve of $\theta_K^M(0.49\,\text{T})$ represents the conventional $M$-linear MOKE in the SkL phase at 0.49 T. **c** Magnetic-field dependence of $\theta_K^T(\mu_0 H_{int})$ normalized by

$\theta_K^M(0.49\,\text{T})$. Because the $\theta_K^M(0.49\,\text{T})$ traverses zero at 0.8 eV and 0.95 eV, $\theta_K^T(\mu_0 H_{int})/\theta_K^M(0.49\,\text{T})$ around these energies tends to diverge in the whole magnetic-field region, and therefore, are not shown here. For more details, see Supplementary Fig. 4. **d** Magnetic-field dependence of $\theta_K^T(\mu_0 H_{int})/\theta_K^M(0.49\,\text{T})$ for 0.15 eV (red), 0.3 eV (blue), 1.1 eV (purple), and 1.7 eV (green). In (**c**) and (**d**), the dotted horizontal lines represent the phase boundaries of the SkL phase.

the topological Kerr rotation spectrum $\theta_K^T(\omega)$ as the deviation of the measured signal from the conventional $M$-linear term[11,15,30]; $\theta_K^T(\omega) = \theta_K(\omega) - \theta_K^M(\omega) = \theta_K(\omega) - \frac{M(\mu_0 H_{int})}{M(2.5\,\text{T})}\theta_K(\omega, 2.5\,\text{T})$, where $\mu_0 H_{int}$ represents the internal magnetic field calibrated by the demagnetization factor, and $M(\mu_0 H_{int})$ and $M(2.5\,\text{T})$ respectively represent the measured magnetization at $\mu_0 H_{int}$ and 2.5 T (see also Methods). The $M$-linear term $\theta_K^M(\omega)$ at each magnetic field is defined as $\theta_K(\omega)$ in the IC-2 phase (2.5 T), with a scaling factor proportional to the normalized magnetization $\frac{M(\mu_0 H_{int})}{M(2.5\,\text{T})}$. This component indeed reproduces the spectra for the IC-1 and IC-2 phases with no net emergent magnetic field (black dotted curves in Fig. 1d). Accordingly, the shaded areas in Fig. 1d express the topological MOKE $\theta_K^T(\omega)$. Figure 2b shows the magnetic-field dependence of $\theta_K^T(\omega)$, demonstrating the appearance of resonance structures around 0.1 eV, 0.35 eV, and 0.6 eV only for the SkL phase (red shaded spectrum). The contour map of the $\theta_K^T(\omega)$ at 0.3 eV, superimposed onto the magnetic phase diagram, also confirms that the emergence of $\theta_K^T(\omega)$ well coincides with the SkL phase (Fig. 1b). These observations establish the topological MOKE emergent from SkL formation. We note that a similar topological magneto-optical

response is observed also for the ellipticity $\eta_K(\omega)$ (Fig. 1e and Supplementary Fig. 3), being connected to $\theta_K(\omega)$ through the Kramers–Kronig relations. Figure 2c shows the magnetic field dependence of $\theta_K^T(\omega)$ normalized by the conventional $\theta_K^M(\omega)$ in the vicinity of the SkL phase (0.49 T), representing the relative strength of the emergent magnetic field as compared to the $M$ at each photon energy. The absolute values for the rotation angle show marked enhancement at 0.15 eV and 0.3 eV; in particular, $\theta_K^T$ is about two times larger than the conventional $M$-induced $\theta_K^M$ in the SkL phase (Fig. 2d, red and blue markers), demonstrating that the topological MOKE is clearly intertwined with the appearance of the SkL.

It should be emphasized that the spectral characteristics of the topological MOKE $\theta_K^T(\omega)$ are totally distinct from those of the conventional $M$-induced $\theta_K^M(\omega)$ (dotted curve, Fig. 2b), suggesting different microscopic origins between these two signals. Resonance structures in $\theta_K^T(\omega)$ are prominent up to 0.8 eV (Fig. 2b, see also Supplementary Fig. 4), manifesting the significant impact of SkL formation on the interband optical transitions, whose energy scale is much higher than the carrier scattering rate (~15 meV; see Supplementary Figs. 5, 6). Thus, it can be concluded that the topological

MOKE is caused by a reconstruction of the electronic bands, including those bands rather far from the Fermi level, through the impact of the emergent field in momentum space that appears in the SkL phase. Note that recent theory predicted that the resonance energy of the topological magneto-optical effect is as large as the exchange interaction[27,31]; the observed resonances might indicate the optical transitions among exchange-split band pairs.

The change in the electronic structure responsible for the topological MOKE is also closely related to the THE in the d.c. limit. To see this, we introduce the optical Hall conductivity for THE, $\sigma_{xy}^T(\omega)$, which is deduced from $\theta_K^T(\omega)$ and the optical conductivity spectra $\sigma_{xx}(\omega)$ (see Methods), enabling a direct comparison with THE. In general, all the resonance structures in the optical Hall conductivity $\sigma_{xy}(\omega)$ contribute, more or less depending on the respective spectral weight and the energy position, to the d.c. Hall response[32–35]. The peak structure in the imaginary part of $\sigma_{xy}(\omega)$, Im $\sigma_{xy}(\omega)$, signals the existence of a resonant optical transition. The spectral weight of Im $\sigma_{xy}(\omega)/\omega$ is equal to the contribution to the d.c. Hall conductivity, according to the sum rule for optical spectra[34]. Hence, Im $\sigma_{xy}(\omega)$ provides an insight into the origin of d.c. Hall conductivity. On the other hand, the real part of $\sigma_{xy}(\omega)$, Re $\sigma_{xy}(\omega)$, shows a dispersive spectral shape close to the resonance, and Re $\sigma_{xy}(\omega = 0)$ must be equal to the Hall conductivity observed in transport measurements. In the present case, $\sigma_{xy}^T(\omega)$ exhibits a pronounced resonance centered around 0.07 eV, with modest features above 0.3 eV (Fig. 3a). The emergence of these resonance structures from interband transitions well coincides with the SkL phase (Fig. 3b, c), suggesting their significant contribution to the d.c. THE. We conclude that the

momentum-space structure of emergent field, or equivalently the Berry curvature, plays an important role for the THE in the SkL phase[36–38]. Note that these energy structures cannot be directly observed by d.c. transport measurements.

A noticeable difference between $\sigma_{xy}^T(\omega)$ and $\theta_K^T(\omega)$ (Fig. 2b) is that the lowest-lying optical transition around 0.07 eV becomes prominent in $\sigma_{xy}^T(\omega)$ (Fig. 3a). This lower-lying resonance with large spectral weight naturally contributes more to the THE than the higher-lying resonances. In the case of the intrinsic anomalous Hall effect of collinear ferromagnets, a low-energy resonance in $\sigma_{xy}(\omega)$, as well as the d.c. response, are often produced by interband optical transitions around avoided band crossing points with intense Berry curvature[32–35], for which spin-orbit coupling is essential. Thus, the presently observed low-energy resonance signals that SkL formation with the scalar spin chirality induces band reconstruction, likely inducing such avoided crossing points accompanied by generation of Berry curvature. Note that the spin-orbit coupling is not essential for the topological MOKE, but may somehow affect the band structure and topological MOKE spectra.

It should be noted that the lowermost-energy value of Re $\sigma_{xy}^T(\omega)$, and the peak value of Im $\sigma_{xy}^T(\omega)$, reach almost 10% of the d.c. topological Hall conductivity ~350 $\Omega^{-1}$cm$^{-1}$ (Fig. 3c). Therefore, the $\sigma_{xy}^T(\omega)$ is expected to be further enhanced below the present experimental window (40 meV – 2.1 eV), suggesting the presence of other optical transitions below 40 meV. Since this energy scale is comparable to that of conduction electron dynamics (~15 meV; Supplementary Fig. 6), a part of this additional contribution may originate from intraband transitions of conduction electrons steeply enhanced below 20 meV,

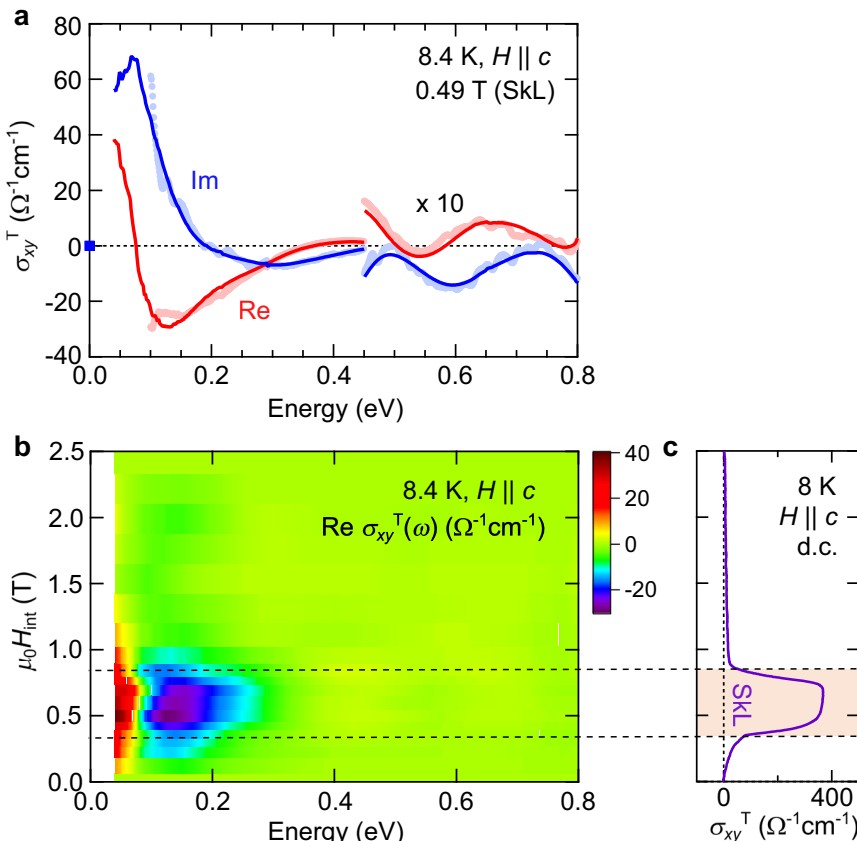

**Fig. 3 | Topological Hall conductivity spectra. a** The topological Hall conductivity spectra $\sigma_{xy}^T(\omega)$ in the SkL phase (0.49 T). The red (blue) markers and curves represent the real (imaginary) parts. The markers show the spectra calculated by using $\theta_K$ and $\eta_K$, measured independently above 0.1 eV, and the curves represent the spectra obtained by the Kramers–Kronig relation. For more details, see Methods. The spectra above 0.45 eV are multiplied by a factor of 10 for clarity. **b** Magnetic-field dependence of the real part of the topological Hall conductivity spectrum, Re $\sigma_{xy}^T(\omega)$, at 8.4 K. **c** Magnetic-field dependence of the d.c. topological Hall conductivity $\sigma_{xy}^T$ at 8 K. In (**b**) and (**c**), the dotted horizontal lines represent the phase boundaries of the SkL phase.

related to the real-space Berry phase (Supplementary Note 1 and Supplementary Fig. 7). Thus, the THE in the present compound probably has two different origins, as suggested by recent calculations[37]. This is in contrast to THE in the conventional B20-type chiral magnets, which is interpreted solely by the emergent magnetic field arising from Berry phases in real space electron motion[11].

The SkL is found to exhibit topological magneto-optical phenomena, which markedly enhance the optical rotation, even without spin-orbit coupling. This suggests a new design principle for large magneto-optical activity that does not rely on heavy elements having large spin-orbit coupling. Besides the enhanced magneto-optical phenomena, the low-energy interband resonance responsible for the THE has the potential to realize a quantized topological magneto-optical effect in certain conditions[24]. The topological MOKE observed here up to the sub-eV near-infrared region exemplifies a strong light-skyrmion interaction, potentially enabling sensitive, noncontact, and fast skyrmion detection, and even optical control of skyrmions, by using commercially available semiconductor diode and fiber lasers. For example, the conventional imaging technique can be applicable to visualize the skyrmion cluster larger than the diffraction limit of light. In addition, several optical techniques that overcome the diffraction limit of light, whose spatial resolution approaches $\lambda/100$[39], might be exploited to detect the response from a single skyrmion particle. Thus, our findings pave a way for novel skyrmion-based devices in conjunction with feasible laser photonics technology.

## Methods

### Single crystal growth and characterization
A single crystal was grown by the optical floating zone technique[14]. The sample was characterized by powder x-ray diffraction (XRD) and energy-dispersive x-ray spectroscopy (EDX). The magnetization was measured in a Magnetic Property Measurement System (Quantum Design). The Hall resistivity was measured by using Physical Property Measurement System (Quantum Design). The magnetic phase diagram was determined from anomalies in the magnetic-field dependence of the magnetization. We corrected the magnitude of magnetic fields within the sample by considering the demagnetization effect. This allows us to compare the transport data, magnetization, and MOKE.

### Magneto-optical Kerr effect measurement
Polar magneto-optical Kerr spectroscopy was performed with use of a Fourier-transform infrared spectrometer (FTIR) for 0.04 – 1.2 eV and a monochromator-type spectrometer for 1.2–2.1 eV. External magnetic fields up to 3 T were applied perpendicular to the sample surface by using a superconducting magnet. For high-precision polarimetry, we used a photo-elastic modulator (PEM) in conjunction with a MCT detector for 0.1–1.2 eV and a Si photodiode above 1.2 eV[40]. The detection of synchronous signal of the reflected light with the fundamental and second harmonic of the modulation frequency enables us to simultaneously measure the Kerr ellipticity $\eta_K$ and the rotation angle $\theta_K$, respectively. To deduce the Kerr spectra, we anti-symmetrized the spectra for the positive and negative magnetic fields. For the measurement in 0.04–0.08 eV, we used a liquid He-cooled bolometer detector and put two wire-grid polarizers before and after the sample, which are oriented at 45 degrees with respect to each other. Only the Kerr rotation angle can be measured by comparing the light intensity at positive field $I(+H)$ and that at negative magnetic field $I(-H)$[41]:

$$\theta_K = \frac{1}{2}\frac{I(+H)-I(-H)}{I(+H)+I(-H)}. \tag{1}$$

### Optical conductivity $\sigma_{xx}(\omega)$ and optical Hall conductivity spectra $\sigma_{xy}(\omega)$
The optical conductivity spectra $\sigma_{xx}(\omega)$ and dielectric constant $\varepsilon_{xx}(\omega)$ were deduced through the Kramers–Kronig transformation of the reflectivity spectra from 0.02 to 40 eV (Supplementary Fig. 5). For extrapolation of the reflectivity data below the lowest energy, we used the Hagen–Rubens relation. The optical Hall conductivity $\sigma_{xy}(\omega)$ can be calculated from the following formula:

$$\sigma_{xy}(\omega) = -\sigma_{xx}(\omega)\sqrt{\varepsilon_{xx}(\omega)}\big(\theta_K(\omega) + i\eta_K(\omega)\big). \tag{2}$$

Here we used zero-field data for the $\sigma_{xx}(\omega)$ and $\varepsilon_{xx}(\omega)$ spectra and we confirmed that these spectra are almost unchanged by the application of a magnetic field (Supplementary Fig. 5). We also note that the $\sigma_{xy}(\omega)$ spectra cannot be directly calculated below 0.1 eV, because the ellipticity $\eta_K(\omega)$ cannot be measured in our present setup in the low-frequency range. Thus, we used $\eta_K(\omega)$ deduced from the Kramers–Kronig analysis of $\theta_K(\omega)$, as described in the next section and in Supplementary Fig. 8. The obtained $\sigma_{xy}(\omega)$ spectra at several magnetic fields are displayed in Supplementary Fig. 9. Since these $\sigma_{xy}(\omega)$ spectra below 40 meV are probably less accurate, due to the suppression of optical rotation caused by the divergence of $\varepsilon_{xx}(\omega)$ in the low-energy region, we focus on the data above 40 meV.

### Kramers–Kronig analysis of magneto-optical spectra
Since the Kerr rotation angle, $\theta_K(\omega)$, is measured down to 40 meV and the Kerr ellipticity, $\eta_K(\omega)$, is not available below 100 meV, we deduce $\eta_K(\omega)$ and the Hall conductivity spectra, $\sigma_{xy}(\omega)$, below 100 meV by the following analysis. We first extrapolate $\theta_K(\omega)$ down to zero energy. The Kerr angle $\theta_K(\omega)$ should converge to zero at zero photon energy, because $\varepsilon_{xx}(\omega)$ shows the $1/\omega^2$ divergence towards zero energy due to the Drude response of conduction electrons (Supplementary Fig. 5d) and $\theta_K(\omega)$ is inversely proportional to $\varepsilon_{xx}^{1/2}$. Thus, we linearly extrapolate $\theta_K(\omega)$ from 40 meV to 0 meV as $\theta_K(\omega) \propto \omega$ (the effect of this extrapolation is discussed in Supplementary Note 2 in more detail). Second, using the extrapolated spectra of $\theta_K(\omega)$ (red solid curves in Supplementary Fig. 8), $\eta_K(\omega)$ was calculated from the Kramers–Kronig relation: $\eta_K(\omega) = -\frac{2\omega}{\pi}P\int_0^\infty \frac{\theta_K(\omega')}{\omega'^2-\omega^2}d\omega'$ (blue solid curves in Supplementary Fig. 8). The resulting $\eta_K(\omega)$ are in quantitative agreement with the $\eta_K(\omega)$ data measured above 100 meV (blue markers in Supplementary Fig. 8), confirming the validity of our Kramers–Kronig analysis.

### Extraction of the optical Hall conductivity spectra for the topological Hall conductivity
We define the contribution to the optical Hall conductivity spectra induced by skyrmion formation as the topological Hall conductivity $\sigma_{xy}^T(\omega)$. This signal is deduced by subtracting the contribution of the anomalous Hall effect from the total $\sigma_{xy}(\omega)$, similar to the case of the d.c. THE[14,15]. The optical Hall conductivity derived from the anomalous Hall effect $\sigma_{xy}^M(\omega)$ is proportional to the magnetization, and therefore, given by

$$\sigma_{xy}^M(\omega,\mu_0 H_{int}) = \frac{M(\mu_0 H_{int})}{M(2.5\,T)}\sigma_{xy}(\omega, 2.5\,T). \tag{3}$$

Eventually, $\sigma_{xy}^T(\omega)$ is described by

$$\sigma_{xy}^T(\omega) = \sigma_{xy}(\omega) - \sigma_{xy}^M(\omega) = \sigma_{xy}(\omega) - \frac{M(\mu_0 H_{int})}{M(2.5\,T)}\sigma_{xy}(\omega, 2.5\,T). \tag{4}$$

We note that, since the magnetic-field variation of $\sigma_{xx}(\omega)$ and $\varepsilon_{xx}(\omega)$ are negligible, this equation can be rewritten as

$$\sigma_{xy}^T(\omega) = -\sigma_{xx}(\omega)\sqrt{\varepsilon_{xx}(\omega)}\big(\theta_K^T(\omega) + i\eta_K^T(\omega)\big). \tag{5}$$

Here we omit the optical Hall conductivity derived from the normal Hall effect due to its small magnitude, shown in Supplementary Note 1. The obtained $\sigma_{xy}^T(\omega)$ indeed shows resonance structures only in the skyrmion phase (Fig. 3), validating our analysis.

## Data availability
All other data that support the plots within this paper are available from the corresponding authors upon reasonable request. The data that support the plots of this study are available from the corresponding author upon reasonable request. Source data are provided with this paper.

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

## Acknowledgements
The authors thank R. Kurokawa and K. Ueda for experimental help and T. Nomoto and L. Spitz for fruitful discussions. This work was partly supported by JST grants JPMJFR212X and JPS KAKENHI grants 21H01796, 22H04470.

## Author contributions
Y. To, Y.O. and Y. Ta conceive the project. Y.D.K. performed the experiment and analyzed data under the supervision of Y.O. and Y.Ta. M.H. prepared the sample. All authors discussed and interpreted the results with inputs from other authors. Y.D.K., Y.O., and Y. Ta wrote the manuscript with the assistance of other authors.

## Competing interests
The authors declare no competing interests.
