## [Peer Review File · Nature Communications]

Reviewers' Comments:

Reviewer #1:

Remarks to the Author:

This work is devoted to an experimental study of the high-frequency (optical) response of a lattice of skyrmions formed in a Gd₂PdSi₃ crystal in a certain range of temperatures and magnetic fields. It seems obvious that a system with a hidden magnetic field has an additional contribution to the magneto-optical response. However, the dependence of this contribution on the frequency of the radiation is nontrivial, and its determination is of considerable scientific interest. In this connection, the nonmonotonic character of the magneto-optical response from the skyrmion lattice at energies of about 0.15 eV, 0.3 and 0.6 eV, discovered by the authors, deserves attention and discussion. From the text of the article it is not clear what this nonmonotonicity is connected with. On the other hand, the theoretical work directly devoted to this problem [Sorn, S., Yang, L. & Paramakanti, A. Resonant optical topological Hall conductivity from skyrmions. Phys. Rev. B 104, 134419 (2021).] indicates that the resonant nature of the magneto-optical response should be associated with the resonant one, corresponding to the spin splitting of subbands. The same conclusion can be drawn from simple semiclassical estimates (M.V. Sapozhnikov et al, Direct observation of topological Hall effect in Co/Pt nanostructured films, Phys. Rev. B 103, 054429 (2021)). In the latter work, it was theoretically and experimentally shown that the magneto-optical response from the skyrmion array is suppressed outside resonances. The authors should give a qualitative explanation of the observed resonances, also taking into account that there are at least two of them (0.3 and 0.6 eV). Are these quantities related to the energy of spin splitting of subbands? In addition, the authors' statement about the possibility of optical probing of skyrmions, due to the difference in their sizes and the wavelength of optical radiation, looks completely inappropriate.

After the corrections and additions, the article can be published.

Reviewer #2:

Remarks to the Author:

Magneto-optical effects (MOE), which reflect the basic interactions between light and magnetism, have been known for more than a century and a half. Recently, several novel findings in this ancient realm have been revealed. Among them, the topological magneto-optical effects (TMOE), originating from the finite scalar spin chirality, were discovered. The TMOE is independent of spin-orbit coupling and band exchange splitting, thus differing from the normal magnetization-induced MOE. This manuscript presents experimental proof of this emergent member of the magneto-optical family in the skyrmion lattice. The major finding is that the skyrmion-type magnetic lattice in Gd₂PdSi₃ gives rise to the TMOE. The results of the current work are novel, interesting, and comprehensive, and the writing is straightforward. This work will most likely inspire more experimental studies on the TMOE in skyrmion lattice and chiral magnets, similar to its dc counterpart - the topological Hall effect. Therefore, I would like to recommend its publication in Nature Communications.

Some minor comments should be considered before publication:

1. The introduction of the TMOE in the manuscript is somewhat misleading. While the statement that "some recent theories predict the so-called topological magneto-optical effect ... (refs. 24, 25), This mechanism does not postulate the relativistic spin-orbit interaction" is technically correct, it should be noted that the term "topological magneto-optical effects" was first proposed in ref. 24, and subsequent work in ref. 25 did not explicitly mention this concept. It is also important to mention that the TMOE can emerge without SOC and band exchange splitting simultaneously. Additionally, the statement that "the low-energy interband resonance responsible for the THE has the potential to realize a quantized topological magneto-optical effect in certain conditions (refs. 24, 35, 37)" is somewhat misleading, as the quantized topological magneto-optical effect was only proposed in ref. 24 and not in refs. 35 and 37.

2. The method used to obtain the topological component of Kerr angle and optical Hall conductivity involves subtracting the magnetism-linear components from the total signals. However, this

method only utilizes data for magnetization M at 2.5 T and assumes a complete linear dependence of M on the magnetic field from 0 to 2.5 T. As shown in Fig. 1c, the slopes of M differ throughout the entire range of magnetic field. Therefore, the topological magneto-optical effects calculated using this method may have significant deviations, particularly at the two boundaries of the SkL phases.

3. The authors could provide further clarification regarding the underlying physics behind the difference in magnitude between the lowermost-energy value of topological optical Hall conductivity and the dc topological Hall conductivity. Specifically, they could discuss how the intra-band transitions of conduction electrons are related to the real-space Berry phase and how this may contribute to the observed difference. Additionally, since the energy range in this work is in the sub-eV region, it may be useful for the authors to comment on the dominant intra-band transition mechanisms in this energy range, such as the Drude term. This could provide further insights into the observed results.

4. The material Gd_2PdSi_3 includes some heavy elements, e.g., Gd and Pd. How to distinguish the contributions from scalar spin chirality and spin-orbital coupling to the TMOE?

5. The authors say that "... d.c. response, are often produced by interband optical transitions around avoided band crossing points with intense Berry curvature, for which spin-orbit coupling is essential. ... SkL formation with the scalar spin chirality induces band reconstruction, likely inducing such avoided crossing points accompanied by generation of Berry curvature." How to confirm whether the avoided crossing points are induced by scalar spin chirality or spin-orbital coupling as the authors also point out that "these energy structures cannot be directly observed by d.c. transport measurements."

6. What would be the effect of canting the magnetic field away from the c-axis or applying it in the xy-plane?

Reviewer #3:

Remarks to the Author:

The manuscript by Kato et al. reported enhanced magneto-optical effect in the sub eV range found in the Gd_2PdSi_3 skyrmion lattice at 8 K, which is then termed as the 'topological magneto-optical effect'. The scientific finding is novel, and the manuscript is well written. I therefore recommend this to be published on Nature Communications providing the following questions can be answered and corrections can be made.

1. Despite having a phase diagram in Figure 1(b) covering a range of temperatures and magnetic fields, later on all spectroscopic data shown in the manuscript are solely based on field variation at 8 K to control the emergence/absence of skyrmions. In order to concretely establish a one-to-one correspondence between the skyrmion lattice and MO phenomena, it would be ideal to also present different spectroscopic data with varying the sample temperatures, to go through the SkL, IC-2 and PM phase, and see if the MO effect resembles the field-dependent data.

2. In Supplementary Figure 7, the polynomial function fits well to the experimental data at 0.13-0.20 eV, however they are substantially different in the 0.10-0.13 eV and 0.04-0.08 eV range. Can the authors comment on the difference? Also, what's the reason of the large standard deviation in the 0.04-0.08 eV data? Further, is there a fundamental reason that a polynomial function can be used to extrapolate the spectroscopic data, is this purely empirical?

3. In Line 131 on Page 4, the authors seem to refer to Supplementary Figure 8 instead of Figure 4. If so, please correct.

1. Reflecting Reviewer #1's comment 1, we discussed the mechanism of topological magneto-optical effect based on the previous theoretical studies (Page 7 line 137).
2. Reflecting Reviewer #1's comment 2, we discussed the possibility of optical probing of skyrmions (Page 9 line 192 and 194).
3. Reflecting Reviewer #2's comment 1, we revised the references in Page 3 line 51 and Page 9 line 190. We also emphasized that the topological magneto-optical effect can emerge without SOC and band exchange splitting simultaneously (Page 3 line 52).
4. Reflecting Reviewer #2's comment 2, we explicitly discussed the analysis procedure (Page 5 line 107).
5. Reflecting Reviewer #2's comment 3, we discussed the relation between intraband transitions and the real-space Berry phase in more detail (Page 9 line 179, Supplementary Note 1, and Supplementary Fig. 7).
6. Reflecting Reviewer #2's comment 4 and 5, we added the comment about the spin-orbit interaction and scalar spin chirality in revised manuscript (Page 8 line 170).
7. Reflecting Reviewer #3's comment 1, we mentioned the temperature dependence (Page 5 line 95) and appended Supplementary Fig. 2.
8. Reflecting Reviewer #3's comment 2, we discussed the validity of the ω -linear function for the low-energy extrapolation in the Method section (Page 12 line 247) and Supplementary Note 2.
9. Reflecting Reviewer #3's comment 3, we newly referred to Supplementary Fig. 6 (Page 7 line 134).

Reply to Reviewer #1's comments

Original report

This work is devoted to an experimental study of the high-frequency (optical) response of a lattice of skyrmions formed in a Gd₂PdSi₃ crystal in a certain range of temperatures and magnetic fields. It seems obvious that a system with a hidden magnetic field has an additional contribution to the magneto-optical response. However, the dependence of this contribution on the frequency of the radiation is nontrivial, and its determination is of considerable scientific interest. In this connection, the nonmonotonic character of the magneto-optical response from the skyrmion lattice at energies of about 0.15 eV, 0.3 and 0.6 eV, discovered by the authors, deserves attention and discussion. From the text of the article it is not clear what this nonmonotonicity is connected with. On the other hand, the theoretical work directly devoted to this problem [Sorn, S., Yang, L. & Paramakanti, A. Resonant optical topological Hall conductivity from skyrmions. Phys. Rev. B 104, 134419 (2021).] indicates that the resonant nature of the magneto-optical response should be associated with the resonant one, corresponding to the spin splitting of subbands. The same conclusion can be drawn from simple semiclassical estimates (M.V. Sapozhnikov et al, Direct observation of topological Hall effect in Co/Pt nanostructured films, Phys. Rev. B 103, 054429 (2021)). In the latter work, it was theoretically and experimentally shown that the magneto-optical response from the skyrmion array is suppressed outside resonances. The authors should give a qualitative explanation of the observed resonances, also taking into account that there are at least two of them (0.3 and 0.6 eV). Are these quantities related to the energy of spin splitting of subbands? In addition, the authors' statement about the possibility of optical probing of skyrmions, due to the difference in their sizes and the wavelength of optical radiation, looks completely inappropriate.

After the corrections and additions, the article can be published.

Our response

We thank Reviewer #1 for spending precious time to review our manuscript and appreciating our work.

Comment 1: In this connection, the nonmonotonic character of the magneto-optical response from the skyrmion lattice at energies of about 0.15 eV, 0.3 and 0.6 eV, discovered by the authors, deserves attention and discussion. From the text of the article it is not clear what this nonmonotonicity is connected with. On the other hand, the

theoretical work directly devoted to this problem [Sorn, S., Yang, L. & Paramakanti, A. Resonant optical topological Hall conductivity from skyrmions. Phys. Rev. B 104, 134419 (2021).] indicates that the resonant nature of the magneto-optical response should be associated with the resonant one, corresponding to the spin splitting of subbands. The same conclusion can be drawn from simple semiclassical estimates (M.V. Sapozhnikov et al, Direct observation of topological Hall effect in Co/Pt nanostructured films, Phys. Rev.B 103, 054429 (2021)) . In the latter work, it was theoretically and experimentally shown that the magneto-optical response from the skyrmion array is suppressed outside resonances. The authors should give a qualitative explanation of the observed resonances, also taking into account that there are at least two of them (0.3 and 0.6 eV). Are these quantities related to the energy of spin splitting of subbands?

Reply 1: Both of the previous theoretical studies raised by the Reviewer suggest that the topological magneto-optical effect is resonantly enhanced at the energy of spin splitting of subbands, i.e., same energy scale with the exchange interaction. According to the predictions, perhaps there are likely to be at least two pairs of subbands, which result in the resonances at 0.3 and 0.6 eV. This theory can phenomenologically explain the observed magneto-optical effect. The correspondence between observed two resonances and the actual band structure in Gd₂PdSi₃ requires more advanced theoretical studies using the ab-initio band calculation including the presence of skyrmion lattice, which is the important future challenge.

We appended this point in the revised manuscript in Page 7 line 137.

Comment 2: In addition, the authors' statement about the possibility of optical probing of skyrmions, due to the difference in their sizes and the wavelength of optical radiation, looks completely inappropriate.

Reply 2: We agree that it is difficult to probe each individual skyrmion particle by the topological magneto-optical effect because of the diffraction limit, if we use the light in free space. Still, it is promising to detect at least skyrmion cluster as large as the wavelength of light by means of the enhanced magneto-optical signal, which is applicable to, for example, the imaging of skyrmion cluster in evaporation and condensation processes. In addition, several optical techniques enable the high-resolution imaging beyond the diffraction limit, as exemplified by scanning near-field optical microscopy with the super-resolution of less than $\lambda/100$ even in infrared region (X. Chen *et al.*, Adv.

Mater. **31**, 1804774 (2019).). Such technological developments may make it possible to optically detect each skyrmion in future.

Reflecting this comment, we discussed the possibility of optical probing of skyrmions in Page 9 line 192 and 194.

Reply to Reviewer #2's comments

Original report

Magneto-optical effects (MOE), which reflect the basic interactions between light and magnetism, have been known for more than a century and a half. Recently, several novel findings in this ancient realm have been revealed. Among them, the topological magneto-optical effects (TMOE), originating from the finite scalar spin chirality, were discovered. The TMOE is independent of spin-orbit coupling and band exchange splitting, thus differing from the normal magnetization-induced MOE. This manuscript presents experimental proof of this emergent member of the magneto-optical family in the skyrmion lattice. The major finding is that the skyrmion-type magnetic lattice in Gd₂PdSi₃ gives rise to the TMOE. The results of the current work are novel, interesting, and comprehensive, and the writing is straightforward. This work will most likely inspire more experimental studies on the TMOE in skyrmion lattice and chiral magnets, similar to its dc counterpart - the topological Hall effect. Therefore, I would like to recommend its publication in Nature Communications.

Some minor comments should be considered before publication:

1. The introduction of the TMOE in the manuscript is somewhat misleading. While the statement that “some recent theories predict the so-called topological magneto-optical effect ... (refs. 24, 25), This mechanism does not postulate the relativistic spin-orbit interaction” is technically correct, it should be noted that the term “topological magneto-optical effects” was first proposed in ref. 24, and subsequent work in ref. 25 did not explicitly mention this concept. It is also important to mention that the TMOE can emerge without SOC and band exchange splitting simultaneously. Additionally, the statement that “the low-energy interband resonance responsible for the THE has the potential to realize a quantized topological magneto-optical effect in certain conditions (refs. 24, 35,

37)” is somewhat misleading, as the quantized topological magneto-optical effect was only proposed in ref. 24 and not in refs. 35 and 37.

2. The method used to obtain the topological component of Kerr angle and optical Hall conductivity involves subtracting the magnetism-linear components from the total signals. However, this method only utilizes data for magnetization M at 2.5 T and assumes a complete linear dependence of M on the magnetic field from 0 to 2.5 T. As shown in Fig. 1c, the slopes of M differ throughout the entire range of magnetic field. Therefore, the topological magneto-optical effects calculated using this method may have significant deviations, particularly at the two boundaries of the SkL phases.

3. The authors could provide further clarification regarding the underlying physics behind the difference in magnitude between the lowermost-energy value of topological optical Hall conductivity and the dc topological Hall conductivity. Specifically, they could discuss how the intra-band transitions of conduction electrons are related to the real-space Berry phase and how this may contribute to the observed difference. Additionally, since the energy range in this work is in the sub-eV region, it may be useful for the authors to comment on the dominant intra-band transition mechanisms in this energy range, such as the Drude term. This could provide further insights into the observed results.

4. The material Gd_2PdSi_3 includes some heavy elements, e.g., Gd and Pd. How to distinguish the contributions from scalar spin chirality and spin-orbital coupling to the TMOE?

5. The authors say that “... d.c. response, are often produced by interband optical transitions around avoided band crossing points with intense Berry curvature, for which spin-orbit coupling is essential. ... SkL formation with the scalar spin chirality induces band reconstruction, likely inducing such avoided crossing points accompanied by generation of Berry curvature.” How to confirm whether the avoided crossing points are induced by scalar spin chirality or spin-orbital coupling as the authors also point out that “these energy structures cannot be directly observed by d.c. transport measurements.”

6. What would be the effect of canting the magnetic field away from the c-axis or applying it in the xy-plane?

Our response

We thank Reviewer #2 for spending precious time to review our manuscript and appreciating our work.

Comment 1: The introduction of the TMOE in the manuscript is somewhat misleading. While the statement that “some recent theories predict the so-called topological magneto-optical effect ... (refs. 24, 25), This mechanism does not postulate the relativistic spin-orbit interaction” is technically correct, it should be noted that the term “topological magneto-optical effects” was first proposed in ref. 24, and subsequent work in ref. 25 did not explicitly mention this concept. It is also important to mention that the TMOE can emerge without SOC and band exchange splitting simultaneously. Additionally, the statement that “the low-energy interband resonance responsible for the THE has the potential to realize a quantized topological magneto-optical effect in certain conditions (refs. 24, 35, 37)” is somewhat misleading, as the quantized topological magneto-optical effect was only proposed in ref. 24 and not in refs. 35 and 37.

Reply 1: Thank you for the thoughtful comments. We revised the references in Page 3 line 51 and Page 9 line 190. We also emphasized that the topological magneto-optical effect can emerge without SOC and concomitant band exchange splitting in Page 3 line 52.

Comment 2: The method used to obtain the topological component of Kerr angle and optical Hall conductivity involves subtracting the magnetism-linear components from the total signals. However, this method only utilizes data for magnetization M at 2.5 T and assumes a complete linear dependence of M on the magnetic field from 0 to 2.5 T. As shown in Fig. 1c, the slopes of M differ throughout the entire range of magnetic field. Therefore, the topological magneto-optical effects calculated using this method may have significant deviations, particularly at the two boundaries of the SkL phases.

Reply 2: In the present analysis, we use the linear interpolation being proportional not to the external magnetic field, but to the measured magnetization at each magnetic field. Specifically, the magnetization-induced component $\theta_K^M(\omega)$ is calculated by $\theta_K^M(\omega) = \frac{M(\mu_0 H_{\text{int}})}{M(2.5 \text{ T})} \theta_K(\omega, 2.5 \text{ T})$, where $M(\mu_0 H_{\text{int}})$ and $M(2.5 \text{ T})$ respectively represent the measured magnetization at $\mu_0 H_{\text{int}}$ and 2.5 T, and $\theta_K(\omega, 2.5 \text{ T})$ represents the Kerr rotation spectrum at 2.5 T. Thus, the complicated field dependence of magnetization

(Fig.1c) is well taken into account, enabling the separation of topological and magnetization-induced components without any assumption. In fact, the Kerr spectrum except for the SkL phase can be explained by the magnetization-linear component, and the topological component is observed only in the SkL phase, as clearly seen in the contour map of topological Kerr rotation (Fig. 1b in the main text).

In the revised manuscript, we explicitly discussed the analysis procedure (Page 5 line 107).

Comment 3: The authors could provide further clarification regarding the underlying physics behind the difference in magnitude between the lowermost-energy value of topological optical Hall conductivity and the dc topological Hall conductivity. Specifically, they could discuss how the intra-band transitions of conduction electrons are related to the real-space Berry phase and how this may contribute to the observed difference. Additionally, since the energy range in this work is in the sub-eV region, it may be useful for the authors to comment on the dominant intra-band transition mechanisms in this energy range, such as the Drude term. This could provide further insights into the observed results.

Reply 3: The real-space Berry phase results in the real-space emergent magnetic field, which induces the electron's traverse motion and topological Hall effect. Since this picture is analogous to the normal Hall effect induced by the Lorentz force, the topological optical Hall conductivity related to the real-space Berry phase can be described by the cyclotron resonance of free carriers under the emergent magnetic field B_{eff} . Accordingly, this intraband contribution $\sigma_{xy}^{T, intra}(\omega)$ is expressed as,

$$\sigma_{xy}^{T, intra}(\omega) = \frac{ne^2}{m^*} \frac{\omega_c^{eff}}{\left(\omega + \frac{i}{\tau}\right)^2 - \omega_c^{eff2}}, \quad (R1)$$

where n is the carrier density, m^* is the effective mass, τ is the scattering time, and $\omega_c^{eff} = eB_{eff}/m^*$ is the cyclotron frequency. To see this spectral response, we assume that the difference between $\sigma_{xy}^T(\omega = 0 \text{ meV})$ and $\sigma_{xy}^T(\omega = 40 \text{ meV})$ comes entirely from the intraband contribution. Figure R1 shows the resultant $\sigma_{xy}^{T, intra}(\omega)$ spectrum at 0.49 T, where we use n , m^* and τ deduced from the transport measurement as discussed in Supplementary Note 1. Here, we put $B_{eff} = 23.5 \text{ T}$. We find that the intraband contribution is steeply enhanced below the present energy window ($\sim 40 \text{ meV}$) and dominates the $\sigma_{xy}(\omega)$ response below 30 meV.

In revised manuscript, we discussed the relation between intraband transitions and the real-space Berry phase in more detail (Page 9 line 179 and Supplementary Note 1), and we showed Fig. R1 in Supplementary Fig. 7.

Fig. R1: The calculated intraband contribution, i.e., topological optical Hall conductivity spectra arising from the real-space Berry phase (green curves). The blue square at zero energy represents the observed d.c. value of the σ_{xy}^T . For comparison, the experimental $\sigma_{xy}^T(\omega)$ (red curves) and $\sigma_{xy}^M(\omega)$ (black curves) dominated by the interband transition are also shown.

Comment 4: The material Gd₂PdSi₃ includes some heavy elements, e.g., Gd and Pd. How to distinguish the contributions from scalar spin chirality and spin-orbital coupling to the TMOE?

Reply 4: The topological magneto-optical (MO) effect arising from scalar spin chirality does not necessarily need the spin-orbit interaction, while the conventional MO effect never exists without spin-orbit coupling. This point is the crucial difference between these two effects, which results in the different magnetization or magnetic-field dependence. Some theoretical works calculated the Hall conductivity under the presence of both the scalar spin chirality and the spin-orbit interaction, and show that the spin-orbit interaction slightly modifies the topological Hall conductivity (R. Ritz et al., Phys. Rev. B 87, 134424 (2013); N. Verma et al., Sci. Adv. 8, eabq2765 (2022)). Thus, the spin-orbit interaction can somehow affect the topological MO effect, although the quantitative estimation of its contribution is difficult at this stage. We also mention that the spin-orbit coupling could be small for the present Gd-based compound because of the small anomalous Hall conductivity directly reflecting the spin-orbit coupling and of the quenched orbital

moment of the Gd-4*f* orbital as discussed in some previous studies (T. Nomoto et al., Phys. Rev. Lett. 125, 117204 (2020); A. Matsui et al., Phys. Rev. B 104, 174432 (2021); S. Spachmann et al., Phys. Rev. B 103, 184424 (2021).).

In conjunction with the Reply 5, we added the comment about the spin-orbit interaction and scalar spin chirality in revised manuscript (Page 8 line 170).

Comment 5: The authors say that "... d.c. response, are often produced by interband optical transitions around avoided band crossing points with intense Berry curvature, for which spin-orbit coupling is essential. ... SkL formation with the scalar spin chirality induces band reconstruction, likely inducing such avoided crossing points accompanied by generation of Berry curvature." How to confirm whether the avoided crossing points are induced by scalar spin chirality or spin-orbital coupling as the authors also point out that "these energy structures cannot be directly observed by d.c. transport measurements."

Reply 5: The spin-orbit interaction alone never give rise to the topological MO effect, and the scalar spin chirality is essential for the emergence of topological MO effect and of the avoided crossing points in skyrmion phase. In this work, we deduced the topological MO spectra based on the field dependence, which provide the information of interband transitions for the avoided crossing points stemming from scalar spin chirality. Meanwhile, the remnant *M*-linear MO effect arises from the interband transition for crossing points solely arising from the spin-orbit interaction.

However, as discussed in Reply 4, the topological MO effect by scalar spin chirality is more or less affected by the spin-orbit coupling in the actual material, so that the corresponding avoided crossing point may be also somehow modified. Therefore, it is difficult to evaluate how the spin-orbit interaction modifies the topological MO effect as well as the avoided crossing points that appear only in skyrmion phase. More precise understanding of electronic band structure under the presence of skyrmion will be an important direction of future study.

In conjunction with the Reply 4, we added the comment about the spin-orbit interaction and scalar spin chirality in revised manuscript (Page 8 line 170).

Comment 6: What would be the effect of canting the magnetic field away from the *c*-axis or applying it in the *xy*-plane?

Reply 6: In the present material, the direction of skyrmion string (or skyrmion tube) is fixed to the c axis due to the uniaxial magnetic anisotropy even when tilting the magnetic field direction from the c axis. According to the previous study (T. Kurumaji et al., Science 365, 914 (2019).), the skyrmion lattice survives up to about $\phi = 45$ deg, where ϕ is the angle between the magnetic field direction and the c axis, as indicated by the Hall signal (Fig. R2). Therefore, when tilting the magnetic field direction from the c axis, the topological magneto-optical effect would be robustly observed up to $\phi = 45$ deg and then abruptly drop to zero, being similar to the red and blue curves in Fig. R2. In stark contrast, the magnetization-induced magneto-optical effect is in proportion to the c -axis component of magnetization and therefore would show the cosine-like behavior with respect to ϕ , being similar to the green curve in Fig. R2.

Fig. R2: Normalized Hall resistivity at 2 K with magnetic field H rotating in the ac plane (the figure is taken from T. Kurumaji et al., Science 365, 914 (2019).). At $\phi = 0$ deg, the SkL and IC-2 phases are stabilized at 9.9 kOe and 40 kOe, respectively. The Hall signal at 9.9 kOe, which is dominated by the topological Hall effect, shows the step-like behavior, suggesting that the skyrmion string direction is fixed to the c axis due to the uniaxial anisotropy and that the SkL suddenly disappears above the certain tilting angle.

Reply to Reviewer #3's comments

Original report

The manuscript by Kato et al. reported enhanced magneto-optical effect in the sub eV range found in the Gd2PdSi3 skyrmion lattice at 8 K, which is then termed as the ‘topological magneto-optical effect’. The scientific finding is novel, and the manuscript

is well written. I therefore recommend this to be published on Nature Communications providing the following questions can be answered and corrections can be made.

1. Despite having a phase diagram in Figure 1(b) covering a range of temperatures and magnetic fields, later on all spectroscopic data shown in the manuscript are solely based on field variation at 8 K to control the emergence/absence of skyrmions. In order to concretely establish a one-to-one correspondence between the skyrmion lattice and MO phenomena, it would be ideal to also present different spectroscopic data with varying the sample temperatures, to go through the SkL, IC-2 and PM phase, and see if the MO effect resembles the field-dependent data.

2. In Supplementary Figure 7, the polynomial function fits well to the experimental data at 0.13-0.20 eV, however they are substantially different in the 0.10-0.13 eV and 0.04-0.08 eV range. Can the authors comment on the difference? Also, what's the reason of the large standard deviation in the 0.04-0.08 eV data? Further, is there a fundamental reason that a polynomial function can be used to extrapolate the spectroscopic data, is this purely empirical?

3. In Line 131 on Page 4, the authors seem to refer to Supplementary Figure 8 instead of Figure 4. If so, please correct.

Our response

We thank Reviewer #3 for spending precious time to review our manuscript and appreciating our work.

Comment 1: Despite having a phase diagram in Figure 1(b) covering a range of temperatures and magnetic fields, later on all spectroscopic data shown in the manuscript are solely based on field variation at 8 K to control the emergence/absence of skyrmions. In order to concretely establish a one-to-one correspondence between the skyrmion lattice and MO phenomena, it would be ideal to also present different spectroscopic data with varying the sample temperatures, to go through the SkL, IC-2 and PM phase, and see if the MO effect resembles the field-dependent data.

Reply 1: Figure R3 shows the temperature dependence of $\theta_{\mathbf{k}}(\omega)$ and $\eta_{\mathbf{k}}(\omega)$, representing the magneto-optical response for the SkL phase, IC-2, and PM phase. The hump around 0.3 eV in $\theta_{\mathbf{k}}(\omega)$ and the dip around 0.2 eV in $\eta_{\mathbf{k}}(\omega)$ are observed in the SkL phase, while

they disappear in the IC-2 and PM phase, being similar to the magnetic-field dependence (Fig. 1d,e in the main text). This observation establishes the topological MO effect also in terms of the temperature dependence.

In revised manuscript, we showed the temperature dependence in Supplementary Fig. 2 and mentioned it in Page 5 line 95.

Fig. R3: Temperature dependence of magneto-optical Kerr (a) rotation angle and (b) ellipticity for $H \parallel c$ at 0.49 T. The data are shifted by vertical offsets of 0.5 mrad. The black dotted curves denote the conventional M -linear MOKE θ_K^M and η_K^M ; the shaded areas represent the topological components θ_K^T and η_K^T .

Comment 2: In Supplementary Figure 7, the polynomial function fits well to the experimental data at 0.13-0.20 eV, however they are substantially different in the 0.10-0.13 eV and 0.04-0.08 eV range. Can the authors comment on the difference? Also, what's the reason of the large standard deviation in the 0.04-0.08 eV data? Further, is there a fundamental reason that a polynomial function can be used to extrapolate the spectroscopic data, is this purely empirical?

Reply 2: We employ two experimental methods for the present measurement depending on the energy range. Above 0.1 eV, the polarization modulation technique can be available, enabling high experimental accuracy. Still, the light intensity tends to be weak in the lower edge of experimental energy window, resulting in worse S/N ratio in the 0.1 – 0.13 eV range. Meanwhile, such the modulation technique cannot be used for 0.04 – 0.08 eV

and we just measure the light intensity by putting two polarizers for polarimetry, so that the standard deviation tends to be large.

In the present Kramers-Kronig analysis, we assume the ω -linear function for the low-energy extrapolation, which is verified in the metallic sample as discussed in the following. In the low-energy region, for example, below 30 or 40 meV, $\epsilon_{xx}(\omega)$ is dominated by the Drude response and shows the $1/\omega^2$ divergence. Therefore, the Kerr rotation and ellipticity, which are given by $-\frac{\sigma_{xy}(\omega)}{\sigma_{xx}(\omega)\sqrt{\epsilon_{xx}(\omega)}}$ and inversely proportional to square root of $\epsilon_{xx}(\omega)$, should be roughly proportional to ω , if we assume the Hall angle $\sigma_{xy}(\omega)/\sigma_{xx}(\omega)$ is constant. We also test some extrapolation functions with similar power-law dependence, which never affects our conclusion.

In the revised manuscript, we discussed the validity of the ω -linear function for the low-energy extrapolation in the Method section (Page 12 line 247) and Supplementary Note 2.

Comment 3: In Line 131 on Page 4, the authors seem to refer to Supplementary Figure 8 instead of Figure 4. If so, please correct.

Reply 3: Thank you for the thoughtful comment. Since the scattering rate is directly discussed in Supplementary Fig. 4, we refer to it in the original manuscript. However, as pointed out, Supplementary Fig. 8 is more convincing to show that the topological magneto-optical effect is induced by the interband transition. Thus, we referred to both figures in the revised manuscript (Page 7 line 134).

Reviewers' Comments:

Reviewer #1:

Remarks to the Author:

I am satisfied with the answers of the authors and recommend publishing this work

Reviewer #2:

Remarks to the Author:

The authors have clarified all my concerns, and I would think the revised manuscript is now ready for publication in Nature Communications.

Reviewer #3:

Remarks to the Author:

The authors have managed to address all my points of concern in the response letter and the revised manuscript. I therefore recommend this manuscript to be published on Nature Communications.

Reply to Reviewer #1's comments

Original report

I am satisfied with the answers of the authors and recommend publishing this work.

Our response

We thank the reviewer for spending precious time to review our manuscript and appreciating our work.

Reply to Reviewer #2's comments

Original report

The authors have clarified all my concerns, and I would think the revised manuscript is now ready for publication in Nature Communications.

Our response

We thank the referee for reviewing our manuscript and evaluating the novelty of topological magneto-optical effect.

Reply to Reviewer #3's comments

Original report

The authors have managed to address all my points of concern in the response letter and the revised manuscript. I therefore recommend this manuscript to be published on Nature Communications.

Our response

We thank the reviewer for his/her positive appraisal and publication recommendation.